# Upper tropospheric CH<sub>4</sub> and N<sub>2</sub>O retrievals from MetOp/IASI within the project MUSICA

Omaira E. García<sup>1</sup>, Eliezer Sepúlveda<sup>2,1</sup>, Matthias Schneider<sup>3</sup>, Andreas Wiegele<sup>3</sup>, Christian Borger<sup>3</sup>, Frank Hase<sup>3</sup>, Sabine Barthlott<sup>3</sup>, Thomas Blumenstock<sup>3</sup>, and Angel M. de Frutos<sup>2</sup>

<sup>1</sup>Izaña Atmospheric Research Centre (IARC), Agencia Estatal de Meteorología (AEMET), Santa Cruz de Tenerife, Spain <sup>2</sup>Atmospheric Optics Group (GOA), University of Valladolid, Valladolid, Spain

<sup>3</sup>Institute of Meteorology and Climate Research (IMK-ASF), Karlsruhe Institute of Technology (KIT), Karlsruhe, Germany

Correspondence to: O.E. García (ogarciar@aemet.es)

Abstract. This paper presents upper tropospheric methane  $(CH_4)$  and nitrous oxide  $(N_2O)$  concentrations retrieved from thermal infrared spectra as observed by the remote sensor IASI (Infrared Atmospheric Sounding Interferometer) on-board the EU-METSAT/MetOp meteorological satellites. The  $CH_4$  and  $N_2O$  mixing ratios are retrieved as side products of the MetOp/IASI retrieval developed for the European Research Council project MUSICA (MUlti-platform remote Sensing of Isotopologues

- 5 for investigating the Cycle of Atmospheric water). The MUSICA/IASI  $CH_4$  and  $N_2O$  retrieval strategy is described in detail as well as their characterisation in terms of the vertical resolution and expected errors. Theoretically, we document that MU-SICA/IASI products can capture the upper tropospheric  $CH_4$  and  $N_2O$  variability (at  $\approx 300-350$  hPa) with a precision better than 2%. We compare the remote sensing data to coincident high precision aircraft vertical profiles taken within the HIAPER Pole-to-Pole Observations (HIPPO) project and empirically estimate a precision of 2.1% (38.2 ppbv) for each individual IASI
- $CH_4$  observation. The precision is improved to 1.7% (32.1 ppbv) for IASI data that have been averaged within  $2^{\circ}x2^{\circ}$  boxes. 10 For N<sub>2</sub>O the empirically estimated precision is 2.7% (8.7 ppbv) for each individual observation and 2.1% (6.9 ppbv) for the  $2^{\circ}x2^{\circ}$  averages. The empirical study works with data from the missions HIPPO1 and HIPPO5, which cover latitudes between  $67^{\circ}$ S and  $80^{\circ}$ N during typical winter and summer conditions in both hemispheres, thus being reasonably representative for global observation during different seasons.
- In addition, we present a product that combines the CH<sub>4</sub> and N<sub>2</sub>O retrieval estimates. The combination is made a-posteriori 15 and we theoretically and empirically show that the combined product has a much better precision than the individual  $CH_4$ and  $N_2O$  products. For the combined product the theoretical precision is 0.8% and the comparison with HIPPO data gives an empirical precision estimate of 1.5% (26.3 ppbv) when considering all individual IASI observations and of 1.2% (21.8 ppbv) for the  $2^{\circ}x2^{\circ}$  averages. In the case that the horizontal, vertical and temporal variation of N<sub>2</sub>O can be robustly modeled, we can easily reconstruct  $CH_4$  from the combined product and generate high quality IASI  $CH_4$  data.
- 20

# 1 Introduction

After carbon dioxide ( $CO_2$ ),  $CH_4$  and  $N_2O$  are currently the most important well-mixed greenhouse gases (GHGs). Although they are much less abundant than  $CO_2$  in the atmosphere, their Global Warming Potentials are significantly larger:  $CH_4$  and

 $N_2O$  are about 35 and 300 times, respectively, more efficient than  $CO_2$  trapping outgoing long wave radiation, on a 100-yr time horizon (Stocker et al., 2013). It is well recognized that the imbalance between their sources and sinks has unquestionably increased during the last few centuries, but the exact location, intensity and nature of  $CH_4$  and  $N_2O$  sources and sinks are not as well understood as those for  $CO_2$  (Crevoisier et al., 2009). The knowledge of today's  $CH_4$  and  $N_2O$  sources/sinks, their

- spatial distribution and their variability in time is essential for understanding their role in the carbon and nitrogen cycles and for a reliable prediction of future atmospheric  $CH_4$  and  $N_2O$  abundances. The latter is important for predicting radiative forcing as well as ozone recovery (both  $CH_4$  and  $N_2O$  act as ozone depleting substances). Existing observations on fluxes of  $CH_4$  and  $N_2O$  from soils and oceans are still insufficient to adequately address these crucial tasks (e.g. Huang et al., 2008; Kort et al., 2011).
- Since the late 1970s, surface in-situ measurements of these GHGs are systematically taken within the GAW programme (Global Atmospheric Watch-World Meteorological Organisation, www.wmo.int). These observations have proved to be very precise and, thus, are indispensable inputs for inverse methods and chemical transport models (e.g. Bousquet et al., 2011; Cressot et al., 2014). More recently, ground-based remote sensing FTS (Fourier Transform Infrared Spectrometer) experiments also routinely provided high-quality CH<sub>4</sub> and N<sub>2</sub>O concentrations in the framework of the international networks
- NDACC (Network for the Detection of Atmospheric Composition Change, www.adc.ucar.edu/irwg, Schneider et al., 2005; Angelbratt et al., 2011; Sepúlveda et al., 2014) and TCCON (Total Carbon Column Observing Network, www.tccon.caltech.edu, Wunch et al., 2011). However, both surface in-situ and ground-based remote sensing measurements sample only a small fraction of the whole atmosphere. In this context, space-based remote sensing instruments have an outstanding importance due to their global coverage, allowing a comprehensive monitoring of the GHGs sources/sinks and their global distributions as well
- as a more complete understanding of the atmospheric processes affecting their flux variations. The great potential of the space-based instruments to observe global CH<sub>4</sub> and N<sub>2</sub>O distributions has extensively been reported in literature. Examples of these satellite measurements by using different spectral ranges and observing geometries are those from ENVISAT/MIPAS (Michelson Interferometer for Passive Atmospheric Sounding, Payan et al., 2009; Plieninger et al., 2015), ENVISAT/SCIAMACHY (SCanning Imaging Absorption SpectroMeter for Atmospheric CHartog-
- raphY, Frankenberg et al., 2006), SCISAT-1/ACE (Atmospheric Chemistry Experiment, De Mazière et al., 2008), AURA/TES (Tropospheric Emission Spectrometer, Wecht et al., 2012; Worden et al., 2012) or GOSAT/TANSO-FTS (Thermal And Near infrared Sensor for carbon Observation, Yokota et al., 2009). Although the thermal nadir instruments have limited sensitivity to the CH<sub>4</sub> and N<sub>2</sub>O concentration variations in the lower troposphere due to the lack of thermal contrast, they have the clear advantage of observing under a large variety of conditions (day and night, over land and ocean, and for partly cloudy
- scenes), increasing significantly their global coverage (Clerbaux et al., 2009; Wecht et al., 2012). Among the current thermal nadir sensors, IASI (Infrared Atmospheric Sounding Interferometer, Blumstein et al., 2004) has special relevance, because it successfully combines the meteorology requirements for weather forecasting (high spatial coverage and a relatively good temporal resolution) and the atmospheric chemistry needs (high spectral resolution thereby allowing for trace gas retrievals), with a long-term data availability. Its mission is guaranteed until 2022 through the meteorological satellites MetOp, the space com-
- ponent of the EUMETSAT (European Organisation for the Exploitation of Meteorological Satellites, www.eumetsat.int) Polar

System (EPS) programme: the first sensor (IASI-A) was launched in October 2006 on-board MetOp-A, the second (IASI-B) was launched in September 2012 on-board MetOp-B and the third (IASI-C) is expected to be launched in October 2018 aboard MetOp-C. As a result of its well-recognized great performance, the IASI mission will be starting in the 2020s with IASI-NG (IASI New Generation, Crevoisier et al., 2014). IASI-NG has a further improved spectral resolution and radiometric perfor-

5 mance and will flown on three successive MetOp-Second Generation (SG) satellites of the EPS-SG system, giving a perspective of data records until the late 2030s. All this is very promising for monitoring atmospheric composition in the long term, and IASI and IASI-NG are key instruments for EUMETSAT's contribution to Copernicus, the European system for monitoring the Earth (e.g. Clerbaux et al., 2009; Crevoisier et al., 2009; August et al., 2012).

The MUSICA/IASI retrieval focuses on tropospheric water vapour isotopologues (Schneider and Hase, 2011; Wiegele et al., 2014; Schneider et al., 2016), but also provides upper tropospheric CH<sub>4</sub> and N<sub>2</sub>O as side products. The presentation of the MUSICA/IASI CH<sub>4</sub> and N<sub>2</sub>O products as well as their empirical validation using the HIAPER Pole-to-Pole Observations (HIPPO) database are the main goals of this paper. In addition, we will use the HIPPO data to empirically validate EUMET-SAT's operational Level 2 (L2) IASI CH<sub>4</sub> and N<sub>2</sub>O products and briefly discuss their quality in comparison to the quality of the MUSICA products.

- The paper is structured as follows: Section 2 presents the MUSICA/IASI  $CH_4$  and  $N_2O$  products (retrieval strategy and theoretical characterisation in terms of vertical sensitivity and error estimation). Section 3 discusses the possibility of combining the  $CH_4$  retrieval data with the co-retrieved  $N_2O$  product and demonstrates that the combined product has theoretically a significantly higher precision than the individual  $CH_4$  and  $N_2O$  products. Section 4 addresses the empirical validation of the MU-SICA/IASI products, detailing the validation dataset and the comparison strategy used as well as the inter-comparison results.
- 20 Section 5 presents the spatial and temporal coverage of the MUSICA/IASI products and very briefly discusses the observed latitudinal gradients and seasonal cycles. Section 6 shows a consistency assessment between the MUSICA/IASI products from the two IASI sensors currently in orbit (MetOp-A/IASI and MetOp-B/IASI). Section 7 briefly extents the validation exercise showing respective results for the EUMETSAT operational L2 IASI CH<sub>4</sub> and N<sub>2</sub>O products. Section 8 summarizes the main results and conclusions of this work.

# 25 2 MUSICA/IASI CH<sub>4</sub> and N<sub>2</sub>O products

## 2.1 MetOp/IASI sensor

30

www.cnes.fr) in cooperation with EUMETSAT. It is on-board the EUMETSAT MetOp satellites, which operate in a polar and low Earth orbit since 2006. With 14 orbits per day in a sun-synchronous orbit (09:30 and 21:30 Local Solar Time equator crossing) IASI can provide global observations twice per day. IASI measures thermal infrared radiation emitted by the Earth's surface and the atmosphere between 645-2760 cm<sup>-1</sup> with an apodized spectral resolution of 0.5 cm<sup>-1</sup> and scans the surface

The IASI sensor is a nadir-viewing Fourier Transform spectrometer developed by CNES (Centre National d'Etudes Spatiales,

for these 120 field of views are disseminated by EUMETSAT as Level 1C (L1C), together with additional information about observation geometry.

# 2.2 CH<sub>4</sub> and N<sub>2</sub>O retrieval strategy

- MUSICA MetOp/IASI retrieval focuses on the optimal estimation of tropospheric water vapour concentrations and on the ratio between the isotopologues HDO and  $H_2O$  (Schneider and Hase, 2011; Wiegele et al., 2014; Schneider et al., 2015). The retrieval analyses the thermal emission spectra recorded by IASI in the 1190-1400 cm<sup>-1</sup> spectral region and uses the thermal nadir retrieval algorithm PROFFIT-nadir (Schneider and Hase, 2011; Wiegele et al., 2014). In the analysed spectral region CH<sub>4</sub> and N<sub>2</sub>O have important spectroscopic signatures and are retrieved simultaneously to the water vapour isotopologues. The PROFFIT-nadir retrieval code has been developed in support of the project MUSICA for analysing thermal nadir spectra.
- It is an extension of the PROFFIT code used since many years for analysing high resolution solar absorption infrared spectra (PROFile Fit, Hase et al., 2004).

The  $CH_4$  and  $N_2O$  VMR (Volume Mixing Ratio) profiles are derived, on a logarithmic scale, using an ad-hoc Tikhonov-Philips slope constraint (TP1 constraint, Tikhonov, 1963) with a strong regularisation. This is almost equivalent to a scaling retrieval and only allows for very small changes in the shape of the a-priori profile. The  $CH_4$  and  $N_2O$  retrievals are made

- simultaneously to retrievals of the water vapour isotopologues as well as to retrievals of the minor interfering species  $CO_2$ and  $HNO_3$ . While the minor interferences of  $CO_2$  and  $HNO_3$  can be well accounted for by scaling the a-priori  $CO_2$  and  $HNO_3$  profiles (scaling retrieval), the interferences of the water vapour isotopologues are very strong and the application of a sophisticated retrieval method is needed. The MUSICA retrieval performs an optimal estimation of isotopologues on a logarithmic scale (Schneider and Hase, 2011; Wiegele et al., 2014).
- A high quality water vapour isotopologue retrieval is crucial for obtaining a  $CH_4$  and  $N_2O$  product with a reasonable quality. This is illustrated in Fig. 1, which shows an example of the radiance measured by IASI and simulated by PROFFIT-nadir in the spectral region used for the  $CH_4$  and  $N_2O$  retrievals as well as the change in IASI radiances due to a change of  $CH_4$  by +5%, of  $N_2O$  by +2%, and of  $H_2O$  by +100%, whereby 5%, 2% and 100% are typical values for the respective trace gas variations (please note the different y-axis scale for  $H_2O$  spectral signatures). As observed, the spectral signatures of  $H_2O$  variations are
- very strong if compared to the signatures of  $CH_4$  and  $N_2O$  variations, meaning that the quality of the  $CH_4$  and  $N_2O$  products depends on a correct interpretation of the spectroscopic interferences of the water vapour isotopologues.

For all the fitted species we use the same a-priori profiles for all retrievals, i.e. they do not vary on a daily, seasonal or latitudinal basis. Thereby, all the observed atmospheric variations are induced by the IASI observations rather than the a-priori information. The a-priori profiles of the different species are typical low-latitudes profiles taken from WACCM (Whole

Atmosphere Community Climate Model-version 5, http://waccm.acd.ucar.edu) provided by NCAR (National Center for Atmospheric Research, J. Hannigan, private communication). They are climatologies provided at a spatial resolution of 1.9°x2.5° and averaged for the 2004-2006 period. The H<sub>2</sub>O isotopologues a-priori data are averages obtained from the isotopologue incorporated global general circulation model LMDZ (Risi et al., 2012).

Together with the gaseous species, surface skin and atmospheric temperatures are also retrieved simultaneously. As a-priori atmospheric temperature profiles, we use the EUMETSAT IASI L2 temperature profiles distributed by the EPS Ground Segment, which are updated for each retrieval. To constrain the a-priori variability, we consider typical variations of 0.25 K for the atmospheric temperature profile, except for the lowermost atmospheric grid point, where variations of 1 K are allowed for. The

5 surface skin temperature retrieval is not constrained.

For the radiative transfer calculations the spectroscopic line parameters are taken from HITRAN 2012 database (Rothman et al., 2013) for all the gases, except for the  $H_2O$  isotopologues. For the latter we use an improved spectroscopy based on HITRAN 2012, but modifying the line intensities (S) for the HDO absorption signatures by +10% (Schneider et al., 2016). This modification is introduced to correct the bias documented in the IASI HDO products reported by Schneider et al. (2015).

- Ocean emissivities are calculated according to Masuda et al. (1988) for three different wavenumbers enveloping the spectral retrieval range, while emissivities at land are taken from the Global Infrared Land Surface Emissivity Database (Seemann et al., 2008) provided as monthly means by the University of Wisconsin in Madison (http://cimss.ssec.wisc.edu/iremis/). The assignation of ground altitude is done using the Global 30 Arc-Second Elevation Dataset (GTOPO30, http://eros.usgs.gov/elevationproducts), in agreement with August et al. (2012). Note that the PROFFIT-nadir retrieval code does not consider the backscatter
- of solar light at the Earth's surface, but this is not critical for simulating the radiances below  $2000 \text{ cm}^{-1}$ .

In this study we only consider cloud-free scenes, based on EUMETSAT L2 cloud products. For details about the EUMET-SAT IASI cloud screening strategy, refer to August et al. (2012) and the Products User Guide (EUM/OPSEPS/MAN/04/0033, EUMETSAT).

# 2.3 Vertical resolution and sensitivity

The vertical structures that are detectable by the IASI sensor are given by the averaging kernel matrix ( $\mathbf{A}$ , avks) obtained in the retrieval procedure. The rows of this matrix describe the altitude regions that mainly contribute to the retrieved target gas VMR profile and, thus, the vertical distribution of the IASI sensitivity. For  $\boldsymbol{x}$  being the actual atmospheric state (the actual trace gas profile) and  $\boldsymbol{x}_{\boldsymbol{a}}$  the a-priori state it is:

$$\hat{\boldsymbol{x}} = \boldsymbol{A}(\boldsymbol{x} - \boldsymbol{x}_a) + \boldsymbol{x}_a + \boldsymbol{x}_\epsilon \tag{1}$$

Here  $\hat{x}$  is the retrieved state and  $x_{\epsilon}$  are the retrieval errors (see Sect. 2.4).

Figure 2 shows the rows of  $\mathbf{A}$  for typical CH<sub>4</sub> and N<sub>2</sub>O observations over ocean pixels in the tropics and polar regions in winter and summer. Since we are applying a strong regularisation on the CH<sub>4</sub> and N<sub>2</sub>O retrieval, the estimated response of IASI to the real atmospheric variability of CH<sub>4</sub> and N<sub>2</sub>O is almost the same for all altitudes, i.e. all the row avks have almost the same shape and peak at almost the same altitude. As observed in Fig. 2, the IASI sensitivity shows a latitudinal

dependency and a weak seasonal dependency. The sensitivity is best in the upper troposphere, i.e.  $\approx 8$  km in polar regions and  $\approx 14$  km in tropics both in summer and winter (intermediate altitudes in middle latitudes, not shown). The full width at half maximum (FWHM) of the row kernels is about 12 km for the tropics and about 10 km for polar latitudes. The kernels indicate

that the MUSICA/IASI retrieval strategy is able to provide  $CH_4$  and  $N_2O$  global distributions of the upper troposphere, which is consistent with the sensitivities obtained by other IASI  $CH_4$  retrievals (e.g. Crevoisier et al., 2013; Xiong et al., 2013).

#### 2.4 Theoretical error estimation

The theoretical error estimations are based on evaluating the error covariance matrices,  $S_{\epsilon}$ , for each uncertainty source considered. Following the formalism given by (Rodgers, 2000),  $S_{\epsilon}$ , is calculated as:

$$\mathbf{S}_{\epsilon} = \mathbf{G}\mathbf{K}_{\mathbf{p}}\mathbf{S}_{\epsilon,\mathbf{p}}\mathbf{K}_{\mathbf{p}}{}^{T}\mathbf{G}^{T}$$
(2)

Here G is the gain matrix sampling the changes in the retrieved VMR profile,  $\hat{x}$ , for changes at the spectral bin y,  $\mathbf{K}_{\mathbf{p}}$  is the parameter Jacobian sampling the changes at the spectral bin y for changes in the parameter p, and  $\mathbf{S}_{\epsilon,\mathbf{p}}$  is the uncertainty covariance matrix for the uncertainty of p. We calculate the parameter Jacobians  $\mathbf{K}_{\mathbf{p}}$  for the error source parameters as listed in Table 1. The calculation consists in simulating two spectra using different values of the parameter p. Then, the differences

10

5

between the two simulated spectra are divided by the difference applied in the parameter p.

The different  $S_{\epsilon,p}$  are given in Table 1. As instrumental errors we consider (i) an conservative IASI radiometric noise of  $2x10^{-2} \mu$ W/cm<sup>2</sup>srcm<sup>-1</sup> (Clerbaux et al., 2009), which corresponds to a measurement noise of 5‰ (noise-to-signal ratio), and (ii) a deviation in the observing geometry (swatch angle) of 0.01 rad. Regarding to model parameters, we assume uncertainties

- for (i) the surface skin temperature (2 K) and atmospheric temperature profile (2 K between 0-2 km, and 1 K above) in agreement with August et al. (2012), (ii) the ground altitude (20 m) since the IASI pixels may cover complex terrain, (iii) the surface emissivity (1%), and (iv) the spectroscopic parameters (line intensity and pressure-broadening parameter) of 2% for CH<sub>4</sub> and N<sub>2</sub>O. For the major interfering species, the water vapour isotopologues, we assume an uncertainty of 1% in the line intensity parameter and an uncertainty of 5% in the pressure broadening parameter. These error values are in concordance with

those reported in the HITRAN database (Rothman et al., 2009). Finally, to account for the humidity interference (so-called cross-dependence on humidity) we assume a variation of 100% of the water vapour isotopologues concentrations.

The error patterns or error vertical profiles are calculated as the square root of the diagonal of the error covariance matrix  $S_{\epsilon}$  for each uncertainty source. Figure 3 shows the estimated error profiles. Because we apply a very strong constraint to the shape of the CH<sub>4</sub> and N<sub>2</sub>O profiles, the errors only weakly depend on the altitude. Note that the total random error is estimated as the root-squares-sum of the measurement noise, the cross dependency on humidity, and all the parameter errors, except for

spectroscopy.

The error budgets for MUSICA/IASI  $CH_4$  and  $N_2O$  products are very similar. The total random error reaches about 2% and is dominated by uncertainties in the atmospheric temperature profile, the measurement noise and spectral interferences with the strong spectral signatures of  $H_2O$  (cross-dependence on humidity). For more humid tropical conditions it can be even by a

30 factor of 1.5 larger (Fig. 3 is representative for a typical mid-latitudinal scenario), which reveals the importance of retrieving the  $CH_4$  and  $N_2O$  observations simultaneously with  $H_2O$ . The spectroscopic parameter uncertainties provide errors of 2%, which is mainly due to the uncertainties in the line intensity parameters of  $CH_4$  and  $N_2O$ . Uncertainties in the spectroscopic parameter of the water vapour isotopologues do not significantly contribute to the  $CH_4$  and  $N_2O$  errors.

#### Dependence on the CH<sub>4</sub> and N<sub>2</sub>O a-priori profiles 2.5

The here presented CH<sub>4</sub> and N<sub>2</sub>O retrievals are made on a logarithmic scale with a very strong constraint on the shape of the profiles and we use a single a-priori for all retrievals. The  $CH_4$  and  $N_2O$  a-priori profiles are shown in Fig. 4(a) as solid cyan and green lines, respectively. If we assume linearity, the a-priori and the averaging kernel fully describe the characteristics of our data. We can assimilate the remote sensing data characteristics to any model data (or vertically resolved in-situ data) by applying the averaging kernel to the model data (or in-situ data) in analogy to Eq. (1):

$$\hat{\boldsymbol{m}} = \boldsymbol{A}(\boldsymbol{m} - \boldsymbol{x}_a) + \boldsymbol{x}_a \tag{3}$$

Here m is the modeled atmospheric state (or the state as measured by vertically high resolving in-situ instruments) and  $\hat{m}$ the smoothed model state (or the smoothed in-situ state) that has the same characteristics as the remote sensing product  $\hat{x}$ . According to Eqs. (1) and (3) we can calculate the smoothed state for any other a-priori state simply by adding  $(\mathbf{A} - \mathbb{I})(\mathbf{x}_a - \mathbb{I})$  $x_{a,\text{new}}$ ) to  $\hat{x}$  and  $\hat{m}$ , whereby  $x_{a,\text{new}}$  is the new a-priori state and  $\mathbb{I}$  an identity matrix.

However, we have to be aware that all the here mentioned operations assume linearity, i.e. it is assumed that A does not depend on  $\hat{x}$ . This is actually not true, because strictly speaking A is calculated for the retrieved state  $\hat{x}$  and might differ for a slightly different  $\hat{x}$ , which would be obtained by using a new a-priori state. In order to estimate the effect of the non-linearities

- we test the effect of changing the a-priori data for a typical mid-latitude and polar retrieval. Please recall that we use a low-15 latitude climatology as single a-priori for all retrievals, so the impact of an inadequate a-priori is statistically larger at middle or high latitudes than at low latitudes. For the typical mid-latitude retrieval we change the a-priori to a mid-latitude climatology (depicted in Fig. 4(a) as dashed blue and dark yellow lines for  $CH_4$  and  $N_2O$ , respectively) and for the typical polar retrieval we change the a-priori to a polar climatology (depicted in Fig. 4(a) as dotted navy blue and dark green lines for  $CH_4$  and
- N<sub>2</sub>O, respectively). We use two methods for determining the retrieval results with the new a-priori: firstly, we assume linearity and add  $(\mathbf{A} - \mathbb{I})(\mathbf{x}_a - \mathbf{x}_{a,\text{new}})$  to the original retrieval states and, secondly, we perform full retrievals using the new a-priori, thereby considering eventual non-linearities. The difference between the two methods is a good estimate for the importance of non-linearities. These differences are depicted in the two right panels of Fig. 4 (panel (b) for the mid-latitude example and panel (c) for the polar example). For the mid-latitudes the differences are smaller than 0.1% throughout the troposphere. For 25 the polar regions it is a bit larger, but still within 0.5% throughout the troposphere.

If we used a varying a-priori we would also need to work with Eq. (3) before comparing remote sensing with model or in-situ data. However, with a varying a-priori we can better consider seasonal or latitudinal climatologies. Then the a-priori state would statistically be closer to the actual atmospheric state. This means that in average  $x - x_a$  (and similarly  $m - x_a$ ) is smaller for a varying a-priori than for a single a-priori, consequently non-linearities would be of less importance (statistically speaking,

because there still might be individual situations where the atmospheric state significantly differs from the climatological state). 30

## 3 Combination of the N<sub>2</sub>O and CH<sub>4</sub> retrieval products

In this section we present a combination of the  $N_2O$  and  $CH_4$  retrieval products, with the final objective to generate a more precise  $CH_4$  product.

# 3.1 Motivation

- 5 When aiming at precise CH<sub>4</sub> observations from space-based platforms, a successful method to reduce CH<sub>4</sub> errors is to combine the retrieved CH<sub>4</sub> observations a-posteriori with the co-retrieved N<sub>2</sub>O estimates (Razavi et al., 2009; Worden et al., 2012). This approach relies on two key issues: (i) CH<sub>4</sub> and N<sub>2</sub>O retrievals similarly behave to many uncertainties sources, such as temperature, clouds and emissivity; and (ii) the atmospheric N<sub>2</sub>O concentrations are rather stable and have continuously grown at an almost constant rate (Stocker et al., 2013).
- 10 In analogy to Eq. (1) we can work with the retrieved  $N_2O$  and  $CH_4$  state vectors in the logarithmic scale and write:

$$\hat{\boldsymbol{x}}_{N_{2}O} = \mathbf{A}_{N_{2}O}(\boldsymbol{x}_{N_{2}O} - \boldsymbol{x}_{a,N_{2}O}) + \boldsymbol{x}_{a,N_{2}O} + \boldsymbol{x}_{\epsilon,N_{2}O}$$

$$\hat{\boldsymbol{x}}_{CH_{4}} = \mathbf{A}_{CH_{4}}(\boldsymbol{x}_{CH_{4}} - \boldsymbol{x}_{a,CH_{4}}) + \boldsymbol{x}_{a,CH_{4}} + \boldsymbol{x}_{\epsilon,CH_{4}}$$
(4)

Here  $\hat{x}$  is the retrieved state vector,  $x_a$  the a-priori state vector, x the state vector that describes the actual atmosphere and  $x_{\epsilon} = \mathbf{GK}_{\mathbf{p}} p_{\epsilon}$  captures the errors due to uncertainties in the retrieval parameters  $p_{\epsilon}$  (for instance, uncertainties in temperature 15 or spectroscopic parameters). As before, the matrices  $\mathbf{A}$  are the averaging kernels.

If we now define the combined product as the difference between the state vectors (difference in the logarithmic scale), we get:

$$\hat{x}_{CH_4} - \hat{x}_{N_2O} = x_{a,CH_4} - x_{a,N_2O} + A_{CH_4}(x_{CH_4} - x_{a,CH_4}) - A_{N_2O}(x_{N_2O} - x_{a,N_2O}) + x_{\epsilon,CH_4} - x_{\epsilon,N_2O}$$
(5)

The idea is that (i) the error of this combined product {x<sub>ε,CH4</sub> - x<sub>ε,N2O</sub>} is much smaller than the errors in the individual
products {x<sub>ε,CH4</sub> or x<sub>ε,N2O</sub>}, and (ii) as N<sub>2</sub>O shares the dynamical variations of CH<sub>4</sub> in the tropopause region, the combined product has a weaker dependency on the tropopause altitude and potentially an improved representativeness of source/sink

#### 3.2 Theoretical treatment of the combined CH<sub>4</sub> product

25

signals.

By a simple matrix multiplication we can make a transformation for the  $\{\ln[N_2O], \ln[CH_4]\}\$  space into the  $\{\frac{\ln[CH_4] + \ln[N_2O]}{2}, \ln[CH_4] - \ln[N_2O]\}\$  space. This transformation between basis systems has been discussed in detail for water vapour isotopologue states in Schneider et al. (2012) and the same approach can be applied for the N<sub>2</sub>O and CH<sub>4</sub> states. The transformation matrix **P** is:

$$\mathbf{P} = \begin{pmatrix} \frac{1}{2}\mathbb{I} & \frac{1}{2}\mathbb{I} \\ -\mathbb{I} & \mathbb{I} \end{pmatrix}$$
(6)

Here, the four matrix blocks have the dimension  $(nol \times nol)$ , and I stands for an identity matrix.

#### 3.2.1 Theoretical error estimation

The error covariance matrix for the transformed states can be easily calculated in analogy to Eq. (2):

$$\mathbf{S}_{\epsilon}' = \mathbf{P}\mathbf{G}\mathbf{K}_{\mathbf{p}}\mathbf{S}_{\epsilon,\mathbf{p}}\mathbf{K}_{\mathbf{p}}^{T}\mathbf{G}^{T}\mathbf{P}^{T} = \begin{pmatrix} \mathbf{S}_{\epsilon11}' & \mathbf{S}_{\epsilon12}' \\ \mathbf{S}_{\epsilon21}' & \mathbf{S}_{\epsilon22}' \end{pmatrix}$$
(7)

There are four matrix blocks with the dimension  $(nol \times nol)$ . The error covariances for the combined product  $\hat{x}_{CH_4} - \hat{x}_{N_2O}$  are collected in the matrix block  $\mathbf{S}'_{\epsilon_{22}}$  (for more details and discussion please refer to Sect. 4.1.2 of Schneider et al., 2012).

The bottom panel of Fig. 3 shows the square root values of the diagonal of the matrix block  $S'_{e22}$  obtained for the different uncertainty sources. It demonstrates that in the combined product the impact of many uncertainty sources is indeed significantly reduced: for atmospheric temperatures, cross dependencies from water vapour, surface temperature, etc. These are the uncertainty sources that are common for N<sub>2</sub>O and CH<sub>4</sub>. The uncertainty due to measurement noise is of course increased, because the measurement noise of N<sub>2</sub>O and CH<sub>4</sub> are independent uncertainty sources and in the combined products the errors are larger than in the individual products. However, the total random error is significantly reduced by a factor of 2 (from 2% to

less than 1%). Measurement noise and uncertainties in the atmospheric temperature profiles are the leading error sources.

In the combined product the error due to uncertainties of spectroscopic parameter is increased, because we assume that the uncertainties in the spectroscopic parameters of  $N_2O$  and  $CH_4$  are independent.

#### 15 3.2.2 Dependence on the a-priori profile

As already mentioned in several sections of the paper we use a single  $N_2O$  and  $CH_4$  a-priori state for all the retrievals, which are averaged from a low-latitude climatology. In particular for higher latitudes this a-priori state might be significantly different from the retrieved state. As a consequence non-linearities can become important. In Sect. 2.5 we describe how the importance of such non-linearities can be estimated for the individual  $N_2O$  and  $CH_4$  products. The respective estimation for the combined product is depicted as navy blue line in Fig. 4 (left panel for mid-latitudes and right panel for polar latitudes). We found that the

20 product is depicted as navy blue line in Fig. 4 (left panel for mid-latitudes and right panel for polar latitudes). We found that the non-linearity can cause misinterpretations in the troposphere of 0.06% and 0.50% for a typical mid-latitude and polar situation, respectively.

## 3.3 Possibility for generating a high quality CH<sub>4</sub> product

Since N<sub>2</sub>O is relatively stable it might be possible that the horizontal and vertical N<sub>2</sub>O distribution is reasonably well captured by atmospheric models. For example, by assimilating high quality N<sub>2</sub>O observations at a few reference locations, it might in the future become possible to generate reliable three dimensional N<sub>2</sub>O fields on a global scale. If  $x_{N_2O}$  is reasonably well know on a global scale, we could add  $A_{N_2O}(x_{N_2O} - x_{a,N_2O}) + x_{a,N_2O}$  to Eq. (5), thereby reconstructing a CH<sub>4</sub> product from the combined product. This reconstructed CH<sub>4</sub> product would have an improved precision (see Fig. 3).