# Peer review of "Upper tropospheric CH4 and N2O retrievals from MetOp/IASI within the project MUSICA"

_Atmospheric Measurement Techniques, 2016_

## Referee Comment (RC1) · Anonymous Referee #1 · 15 Feb 2017

General comments :

The paper is well structured and detailed effort has been given to the underlying theory and error estimation.

However I find that the validation exercise is based on a limited set of collocated data (36 collocations). Why are the HIPPO 2 to 4 profiles not included as well?

In addition, large differences are found between the retrieved CH4 concentrations and the 'CH4 combined with N2O' product which are not well explained. For CH4, you find CH4 vmrs higher than 1900 ppbv, and values around 2000 ppbv are not uncommon. These are issues which are not well addressed.

Specific comments :

P2, L29 : ...partly cloudy scenes... I would not make that statement. You only consider cloud-free scenes as you mentioned in Section 2.2 (P5, L16).

P4, L32 : You use a WACCM climatology averaged for the 2004-2006 period as an a priori set. How do you account for the strong trend in CH4 and N2O of the last decade ?

P5, L5 : What do you mean with 'the surface skin temperature retrieval is not constrained' ?

P5, L16: Is line mixing included ? In the paper of Razavi et al. they investigated the influence of line mixing in the $\nu4$ spectral band around 1300 cm-1 range. They proposed to use the reduced spectral range of 1240-1290 cm-1 for the retrieval of CH4 to reduce the effect of line mixing. How do you account for the effect of line mixing ?

P5, L26: The row of the A at which altitude ?

P5, L29: It would be very interesting to see the complete A matrix (not just 1 row), to have a complete view on the sensitivity of the retrieved product.

P5, L31 : You give an estimate of the vertical resolution of around 12 km in the tropics and around 10 km in the polar regions. But what are typical values for Degrees of Freedom for Signal (DOFS) for the retrievals, as calculated from the trace of the Averaging Kernel ?

P7 : The added value of this section is not clear to me. Is that something you would you like to implement, i.e. to calculate the smoothed state for another a priori state to account for the indeed very different shape of the CH4 and N2O profile for low-, mid- and high-latitudes ?

P9 : Theoretical error estimation / Figure 3 : You give an estimate of the theoretical error for the combine product, but for the 'CH4 combined with N2O'-product you add xN2o (=xa,N2O for simplicity), as given in Equation 8. The error on xa,N2O still needs to be characterized and taken into account. Especially since the xa,N2O is a climatology averaged for 2004-2006 dataset is used which is an underestimation of the current N2O concentrations.

P10, L16 : Why did you not include HIPPO 2, 3 and 4 ? Then you have a much larger dataset to compare. Here you compare only about 2 months of data, 36 collocations. Do you get similar results including HIPPO 2-4 ?

P11, L29: Please add the number of collocated measurements for these 88 valid aircraft profiles.

P11, L31 / Figure 6 : It seems you have quite some variability in the MUSICA N2O and CH4 concentrations (Figures a and b) especially for the latitudes 20-60° S. With unrealistic values (CH4 concentrations > 2000 ppbv). Could you consider an a-posteriori filtering to exclude these outliers ? What are the main differences in the MUSICA CH4 retrievals between these 2 periods (Jan 2009 for HIPPO1, Aug/Sep 2011 for HIPPO5). Do the IASI L2 temperature profiles which you use as a priori for the temperature retrievals have an improved version for the year 2011 (HIPPO 5) ?

P13, L9 : 'The well known seasonal cycle with higher concentrations in summer than in winter'. The seasonal cycle of methane as found by flask measurements (the GAW network) is reversed, with high concentrations in winter and low concentrations in summer. How do you explain this reversed cycle at 350-300 hPa ?

P13, L13 / Figure 8 : Could you show both CH4 and 'CH4 combined with N2O' with the same colorscale ? For CH4 you have values in the range of 1900-1950 ppbv in the Northern Hemisphere for August, which are 'reduced' to values of around 1750 ppbv for the combined product. How do you explain these large differences ? Could you show a difference plot between the 2 products and discuss this ?

P13, L26 : I am surprised by the large scatter between the METOP-A and METOP-B N2O and CH4 retrievals. What are the values for the mean difference and standard deviation of the difference between the IASI-A and IASI-B retrievals for the 3 products

?

P14, L22 : What did you use as the actual state for N2O (or XN2O) ?

Technical comments :

P1, L13 : please add, 'over the Pacific region'.

P2, L18 : ...sample only a small fraction of the whole atmosphere... When you write it like that I would think you mean a reduced altitude range of the atmosphere. Maybe put ...have limited geographical coverage.. ?

P3, L5 : will be flown

P10 , L9 : Empirical validation. You use it throughout the paper. To my knowledge, empirical means without using any scientific method or theory. This validation study seems to be based on a scientific theory, not ? I would omit empirical throughout the paper. (P1, L9; P 3, L 11; P 3, L 12; P12, L18 ;..)

---

## Referee Comment (RC2) · Anonymous Referee #3 · 8 Jun 2017

The paper written by Garcia et al. entitled "Upper tropospheric CH4 and N2O retrievals from MetOp/IASI within the project MUSICA" shows the retrivals of CH4 and N2O using the MUSICA methodology, which is based in Optimal Estimation (alas Rodgers 2000). It is an interesting paper showing how to perform retrivals of these two species for infrared hyperspectral instruments (IASI). There are not many of these kind of retrivals, and a new one based in OE, where uncertainties can be traced back very well, is very welcome. The paper deserves to be published. I would strongly encourage the authors to polish the paper and publish it.

There are a few minor corrections to be done, mainly due to the convoluted way of explaining the subject. Explanations should be made in short sentences, explaining the details. The sentences should come one after the other following a smooth reasoning

and deduction process.

There is also one major correction dealing with what is called in the paper "combined retrieval". It looks as if (difficult to know because I could not follow the explanations well enough, see minor corrections above) two retrievals are done one after the other using exactly the same measurements. This is something that should be totally avoided in OE. Looking at the final uncertainty of the retrievals, Sr, in OE, it is $Sr^{-1} = K^{T} Se^{-1} K + Sa^{-1}$ where Se is the measurement uncertainty covariance matrix and Sa the background uncertainty covariance matrix, and K is the Jacobian. It can easily be seen that Sr ends up being smaller than Sa. For example, for a simple 1D case, if $K^{T} Se^{-1} = 1$ and Sa= 1 then Sr = 0.5. If we now apply OE to the same measurents using this retrieval as a new background (now Sa would be 0.5), we obtain a new Sr=0.333. Much smaller than the correct value obtained initially (0.5). This is because the background or a priori should be information that is completely independent of the measurements, otherwise we are making a big mistake using the OE theory. Because of this, it should be clarified if the "combined retrieval" is this kind of incorrect double OE retrieval or something else.

More especifically:

- Page 4 line 7 insert , - Intro: the biggest greenhouse gas is water vapour. The biggest greenhouse gas which produces climate forcing is CO2. Please include "as climate forcing gases" in this sentence about greenhouse gases. - First paragraph of section 2.4 is not clear. What is exactly s_epsilon, and S_epsilon,p? A priori and posteriori error covariance matrices? What is p, a trace gas, eg, ch4, ... Perhaps a small introduction to OE using tge cost function formula, which is known by everybody, would clarify the notation at the beggining. - Swach should probably be swath

- Section 2.4 is written for a person who already knows the música retrieval. This is not a good way to engage readers. It should be written starting from basic or theory (Rodgers). Perhaps an equation showing what is minimized ( cost function) would add

clarity to the notation used. Likewise, the exact tikonov regularization could be written in a formula - Page 7 line 15. It is well known A changes with profiles. No need to say first we assume linearity and then say it is not true. Jump directly into non-linearity and then, if needed, approximate it to something more or less linear - Last paragraph page 7 very confusing

- Section 3. It is not clear when you do the combined retrieval if you are doing the retrieval twice with the same measurements. Please explain clearly. If this is the case, care should be taken not to use the same measurements twice. Otherwise we will estimate a much smaller error than the real value. See comments about this above. - Section 3.3 confusing - Eq 9 not well explained, probably because combined retrieval not well explained

- Section 4.2. usually time/space collocation windows are chosen with a criteria of little variability in this window. Is this the case here? Is this justified by any paper? If not, why is this particular collocation window chosen ? Reference? - Section 8 line 5 one before last paragraph. Again, combined retrieval seems to mean retrieving the same quatinty with the same measurements using oe. You will get a wrong error covariance matrix like this.

---

## Referee Comment (RC3) · Anonymous Referee #4 · 13 Jun 2017

Overall it is good and can be published after major revision. The major problem is that it is like a report, and many places should be rewritten to make it more concise. For example, the conclusion should be rewritten completely, and the first paragraph in section 8 should be removed.

For structure, sections 5,6,7,9 should be moved to the data or method part, and move the results in these sections to validation, or put some of them to a subsection in validation.

There are quite a lot grammar problems and need some efforts to polish the language.

References should include AIRS which has the similar CH4 and N2O products.

To say IASI as thermal "nadir" sensor is inappropriate. It scans and have a large swath.

[Figure]

The whole paper has not mentioned the quality control of the products, and is the 2X2 average use all retrievals ? I think the retrievals are made on for clear pixels, and it should be mentioned.

Why the HIPPO -2,-3,-4 data are not used in the validation ? How many profiles have been used ?

---

## Author Comment (AC1) · 31 Jan 2018

Dear referees,

We greatly appreciate your collaboration and your comments on our manuscript. We have elaborated replies to your comments and would like to sincerely apologize for the delay. Many thanks for your patience.

First, we would like to remark that the paper has been extensively revised in some key aspects. On the one hand, the MUSICA/IASI CH4 and N2O retrieval strategy has been significantly improved by including (1) a profiling retrieval of CH4 and N2O that capture better the middle and upper tropospheric CH4 and N2O concentration variabilities, (2) the contribution of the water continuum that allow us to better simulate the observed IASI signals, and (3) a better usage of the EUMETSAT L2 cloud flag.

On the other hand, one of the main concerns from the three Referees is the lack of a comprehensive validation database. Therefore, all the improvements introduced by the new version of MUSICA/IASI strategy have been validated with a multi-platform validation database. Together with the HIPPO aircraft profiles used in the first version of the paper (missions 1 and 5), the revised manuscript adds the comparison with (1) the reminder HIPPO missions (2, 3 and 4), (2) continuous in-situ CH4 and N2O observations performed at the subtropical Izaña Atmospheric Observatory (Tenerife, Spain) in the framework of the WMO/GAW programme, and (3) continuous ground-based Fourier Transform Infrared (FTIR) measurements taken in the framework of the NDACC (Network for Detection of Atmospheric Composition Change) at the subtropical Izaña Observatory, mid-latitude Karlsruhe station and polar Kiruna site. This complete validation exercise allow us to document that the MUSICA IASI CH4 products properly distinguish the different annual cycle of the lower/middle troposphere and the upper troposphere. An example demonstration of this profiling capability is shown in the Figure below. It compares the MUSICA IASI products with WMO/GAW in-situ data and ground-based FTIR remote sensing data for different altitudes.

[Figure]

Figure: Annual cycle of the N2O, CH4 and the a posteriori corrected CH4 products for the altitude of 4.2 km (upper panel) and 10 km (bottom panel) as observed by the WMO/GAW in-situ analysers and ground-based FTIR located at the Izaña Atmospheric Observatory and coincident IASI observations.

In the following all the specific comments from each Referee will be addressed separately (Referees' text in italic, authors' reply indented and blue).

Best regards,
Omaira García et al.,

**Anonymous Referee #1**

*General comments :*

*The paper is well structured and detailed effort has been given to the underlying theory and error estimation. However I find that the validation exercise is based on a limited set of collocated data (36 collocations). Why are the HIPPO 2 to 4 profiles not included as well? In addition, large differences are found between the retrieved CH4 concentrations and the 'CH4 combined with N2O' product which are not well explained. For CH4, you find CH4 vmrs higher than 1900 ppbv, and values around 2000 ppbv are not uncommon. These are issues which are not well addressed.*

As aforementioned, the validation database has been significantly extended including all the HIPPO missions.
Specifically, the unrealistic CH4 and N2O values observed likely correspond to pixels partly cloudy (unrecognised clouds by the cloud filter used in the first version of MUSICA retrieval). This has been corrected with the improved cloud filter in the revised manuscript.

Specific comments :

*P2, L29 : ...partly cloudy scenes... I would not make that statement. You only consider cloud-free scenes as you mentioned in Section 2.2 (P5, L16).*

This statement has been modified in the revised manuscript.

*P4, L32 : You use a WACCM climatology averaged for the 2004-2006 period as an a priori set. How do you account for the strong trend in CH4 and N2O of the last decade?*

To compute the climatology used as apriori information we use a reference period where, specially, the CH4 concentrations were stable. This unique apriori is used for all the IASI retrievals, so that we guarantee that all the observed atmospheric variations are induced by the IASI observations.
The comparison at different time scales with independent observations, such as the WMO/GAW in-situ records performed at the Izaña Atmospheric Observatory (Tenerife, Spain) and NDACC ground-based FTIR instruments, document that IASI is able to capture the long-term trends of CH4 and N2O.

*P5, L5 : What do you mean with 'the surface skin temperature retrieval is not constrained' ?*

The surface skin temperature is fitted during the retrieval process without constraint.

*P5, L16: Is line mixing included ? In the paper of Razavi et al. they investigated the influence of line mixing in the _4 spectral band around 1300 cm-1 range. They proposed to use the reduced spectral range of 1240-1290 cm-1 for the retrieval of CH4 to reduce the effect of line mixing. How do you account for the effect of line mixing ?*

We do not account for the effect of line mixing and can comment this in the final manuscript version.

*P5, L26: The row of the A at which altitude?*

Figure 2 displays all the row averaging kernels at the altitude grid used for radiative transfer simulations (28 levels between sea level and 55 km), where the coloured lines represent the row kernels at the IASI maximum sensitivity: 8 km and 14 km for polar and tropics, respectively. Please note that since the CH4 and N2O products are estimated by performing a profile retrieval but with a very strong regularisation (almost equivalent to a scaling retrieval), all the averaging kernels show almost identical shape and peak at the same altitude.

Please note that for the revised manuscript the retrieval scheme has been improved and the constraints reduced. We now perform profile retrievals. The revised manuscript will contain a Figure showing all the entries of a typical averaging kernel.

*P5, L29: It would be very interesting to see the complete A matrix (not just 1 row), to have a complete view on the sensitivity of the retrieved product.*

See comment above.

*P5, L31 : You give an estimate of the vertical resolution of around 12 km in the tropics and around 10 km in the polar regions. But what are typical values for Degrees of Freedom for Signal (DOFS) for the retrievals, as calculated from the trace of the Averaging kernel ?*

As aforementioned, we were performing a profile retrieval but with a very strong regularisation, thereby the DOFs values were very close to unity.

With the new retrieval strategy the DOFs obtained varies from about 1.2 to 1.4 at higher and tropical latitudes, respectively, for N2O and from 1.4 to 1.8 at higher latitudes and tropical latitudes for CH4.

*P7 : The added value of this section is not clear to me. Is that something you would you like to implement, i.e. to calculate the smoothed state for another a priori state to account for the indeed very different shape of the CH4 and N2O profile for low-, mid and high-latitudes ?*

This section aims to analyse the effect of using unique a priori information for all IASI retrievals (independently of latitude or season). However, the added value of this section seems to be limited; thereby this will be removed in final manuscript.

*P9 : Theoretical error estimation / Figure 3 : You give an estimate of the theoretical error for the combine product, but for the 'CH4 combined with N2O'-product you add xN2o (=xa,N2O for simplicity), as given in Equation 8. The error on xa,N2O still needs to be characterized and taken into account. Especially since the xa,N2O is a climatology averaged for 2004-2006 dataset is used which is an underestimation of the current N2O concentrations.*

The combined product is generated a posteriori from the retrieved N2O and CH4 products. Like the N2O and CH4 product the combined product depends on the assumed CH4 and N2O a priori. In the paper we estimate the errors and perform validation studies for the combined product.

We can reconstruct a CH4 product from the combined product. To do so we need to know the accurate N2O variations (from a model or an assimilation product). The error of this reconstructed product is the sum of the error of the combined product and the error in the N2O model data used

for the reconstruction. If accurate models or reanalysis data of N2O are available the reconstructed CH4 product will be of a better quality than the original CH4 product, because the combined product has smaller errors than the CH4 product.

The differences between the combined product and the CH4 product reconstructed from the combined product will be better explained in the revised manuscript.

*P10, L16 : Why did you not include HIPPO 2, 3 and 4 ? Then you have a much larger dataset to compare. Here you compare only about 2 months of data, 36 collocations. Do you get similar results including HIPPO 2-4 ?*

In the revised paper we use all HIPPO profiles from all 5 HIPPO missions to validate the MUSICA/IASI CH4 and N2O products in the middle and upper troposphere. For the middle troposphere, we consider all the HIPPO profiles with measurements covering at least the 2000-8000m altitude range (441 different profiles) and we compare these profiles to all IASI measurements that have sufficient sensitivity for a layer around 4km. For the upper troposphere, we compare IASI data that are sensitive for a layer around 10km altitude. Therefore, we require HIPPO profiles that reach at least 12.5km altitude. There are 22 different HIPPO profiles that we can use for a meaningful comparison to IASI CH4 and N2O retrievals at 10km.

The comparison results including HIPPO 2-4 are very similar. There are no a specific pattern depending on the HIPPO mission.

*P11, L29: Please add the number of collocated measurements for these 88 valid aircraft profiles.*

The final number of collocated HIPPO measurements has been included in the revised manuscript.

*P11, L31 / Figure 6 : It seems you have quite some variability in the MUSICA N2O and CH4 concentrations (Figures a and b) especially for the latitudes 20-60_ S. With unrealistic values (CH4 concentrations > 2000 ppbv). Could you consider an a-posteriori filtering to exclude these outliers ? What are the main differences in the MUSICA CH4 retrievals between these 2 periods (Jan 2009 for HIPPO1, Aug/Sep 2011 for HIPPO5). Do the IASI L2 temperature profiles which you use as a priori for the temperature retrievals have an improved version for the year 2011 (HIPPO 5) ?*

The unrealistic CH4 and N2O values displayed in Figure 6 likely correspond to pixels partly cloudy (unrecognised clouds by the cloud filter used). This has been corrected with the improved cloud filter in the revised manuscript.

As the referee points out, the improvement of the atmospheric temperature profiles (used here as a priori for the temperature retrievals) when changing from EUMETSAT L2 PPF v4 to v5 can have an effect on the long-term behaviour of the MUSICA IASI product. We will briefly discuss this in the revised manuscript.

*P13, L9 : 'The well known seasonal cycle with higher concentrations in summer than in winter'. The seasonal cycle of methane as found by flask measurements (the GAW network) is reversed, with high concentrations in winter and low concentrations in summer. How do you explain this reversed cycle at 350-300 hPa ?*

At 350-300 hPa IASI is only able to follow the annual shift of the tropopause, thus showing higher concentrations in summer than in winter. This seasonal pattern does not coincide with the CH4 seasonal cycle in the lower troposphere, as reflected the continuous in-situ records taken in the framework of the WMO/GAW programme.

With the new version of the MUSICA retrieval, it will be documented that IASI is able to distinguish the different annual cycle of the lower/middle troposphere and the upper troposphere, when comparing with WMO/GAW in-situ records performed at the Izaña Atmospheric Observatory (Tenerife, Spain) and ground-based FTIR instruments (see figure above).

*P13, L13 / Figure 8 : Could you show both CH4 and 'CH4 combined with N2O' with the same colorscale ? For CH4 you have values in the range of 1900-1950 ppbv in the Northern Hemisphere for August, which are 'reduced' to values of around 1750 ppbv for the combined product. How do you explain these large differences ? Could you show a difference plot between the 2 products and discuss this ?*

In the combined product errors that are common to CH4 and N2O are removed. This is mainly temperature uncertainties or effects of clouds, which are likely responsible for the unrealistic values observed in Figure 8. On the other hand systematic errors might be increase, because the systematic errors due to spectroscopic parameters of CH4 and N2O can sum up. All this will be better discussed in the error estimation section of the revised paper.

*P13, L26 : I am surprised by the large scatter between the METOP-A and METOP-B N2O and CH4 retrievals. What are the values for the mean difference and standard deviation of the difference between the IASI-A and IASI-B retrievals for the 3 products?*

This large scatter is mainly due to unrecognised clouds. Cloud filter has been substantially improved in the revised manuscript.

*P14, L22 : What did you use as the actual state for N2O (or XN2O) ?*

Since EUMETSAT L2 CH4 and N2O products are retrieved using Artificial Neuronal Networks, they do not have a priori information. So, the combined EUMETSAT XCH4 products is computed according to Eq . (8) and considering the mean XN2O value of all IASI coincidences as a priori.

*Technical comments :*

*P1, L13 : please add, 'over the Pacific region'.*

*P2, L18 : ...sample only a small fraction of the whole atmosphere... When you write it like that I would think you mean a reduced altitude range of the atmosphere. Maybe put ...have limited geographical coverage.. ?*

*P3, L5 : will be flown*

*P10 , L9 : Empirical validation. You use it throughout the paper. To my knowledge, empirical means without using any scientific method or theory. This validation study seems to be based on a scientific theory, not ? I would omit empirical throughout the paper. (P1, L9; P 3, L 11; P 3, L 12; P12, L18 ;..)*

All the technical comments have been incorporated into the revised manuscript.

_Anonymous Referee #3_

_The paper written by Garcia et al. entitled "Upper tropospheric CH4 and N2O retrievals from MetOp/IASI within the project MUSICA" shows the retrivals of CH4 and N2O using the MUSICA methodology, which is based in Optimal Estimation (alas Rodgers 2000). It is an interesting paper showing how to perform retrievals of these two species for infrared hyperspectral instruments (IASI). There are not many of these kind of retrievals, and a new one based in OE, where uncertainties can be traced back very well, is very welcome. The paper deserves to be published. I would strongly encourage the authors to polish the paper and publish it._

_There are a few minor corrections to be done, mainly due to the convoluted way of explaining the subject. Explanations should be made in short sentences, explaining the details. The sentences should come one after the other following a smooth reasoning and deduction process._

_There is also one major correction dealing with what is called in the paper "combined retrieval". It looks as if (difficult to know because I could not follow the explanations well enough, see minor corrections above) two retrievals are done one after the other using exactly the same measurements. This is something that should be totally avoided in OE. Looking at the final uncertainty of the retrievals, Sr, in OE, it is $Sr^{-1} = K^{T} Se^{-1} K + Sa^{-1}$ where Se is the measurement uncertainty covariance matrix and Sa the background uncertainty covariance matrix, and K is the Jacobian. It can easily be seen that Sr ends up being smaller than Sa. For example, for a simple 1D case, if $K^{T} Se^{-1} = 1$ and Sa= 1 then Sr = 0.5. If we now apply OE to the same measurents using this retrieval as a new background (now Sa would be 0.5), we obtain a new Sr=0.333. Much smaller than the correct value obtained initially (0.5). This is because the background or a priori should be information that is completely independent of the measurements, otherwise we are making a big mistake using the OE theory. Because of this, it should be clarified if the "combined retrieval" is this kind of incorrect double OE retrieval or something else._

In the revised paper we will improve the usage and description of the terminologies "simultaneous retrieval", "combined retrieval products" and "a posteriori corrected CH4 product". Basically, the MUSICA/IASI retrieval strategy offers three products:
  (1) "Simultaneous retrieval of CH4 and N2O": The CH4 and N2O products are retrieved simultaneously with other interfering species and the atmospheric temperature profiles in one unique retrieval process.
Then, a posteriori, we perform some transformations, obtaining:
  (2) "Combined product": a posteriori difference between the retrieved CH4 and N2O products (step 1) according to Eq. (5) in logarithm scale.
  (3) "A posteriori corrected CH4 product": It is obtained summing the "combined product" and accurate N2O simulations (e.g. from model or reanaylses).

_More especifically:_

_- Page 4 line 7 insert , - Intro: the biggest greenhouse gas is water vapour. The biggest greenhouse gas which produces climate forcing is CO2. Please include "as climate forcing gases" in this sentence about greenhouse gases._

This has been corrected in the revised manuscript.

*- First paragraph of section 2.4 is not clear. What is exactly s_epsilon, and S_epsilon,p? A priori and posteriori error covariance matrices? What is p, a trace gas, eg, ch4, ...*

$S_\varepsilon$ is the error covariance matrix due to errors in the forward model parameters. This error covariance is computed for each error source/model parameter considered *p* (eg, spectroscopy, temperature, etc), using the uncertainty covariance matrix of the corresponding model parameter analysed, *p* ($S_{\varepsilon,\,p}$). This will be clarify in the revised manuscript. We will also state that we use a notation according to Rodgers (2000),

*- Perhaps a small introduction to OE using tge cost function formula, which is known by everybody, would clarify the notation at the beggining.*

In the revised manuscript an introduction in Optimal Estimation formulism will be given: notation, equations, etc, to help the readers.

*- Swach should probably be swath*

This has been corrected in the revised manuscript.

*- Section 2.4 is written for a person who already knows the música retrieval. This is not a good way to engage readers. It should be written starting from basic or theory (Rodgers). Perhaps an equation showing what is minimized ( cost function) would add clarity to the notation used. Likewise, the exact tikonov regularization could be written in a formula*

As aforementioned, in the revised manuscript an introduction in Optimal Estimation formulism will be given: notation, equations, etc, to help the readers.

*- Page 7 line 15. It is well known A changes with profiles. No need to say first we assume linearity and then say it is not true. Jump directly into non-linearity and then, if needed, approximate it to something more or less linear*

*- Last paragraph page 7 very confusing*

This section aims to analyse the effect of using unique a priori information for all IASI retrievals (independently of latitude or season). However, the added value of this section seems to be limited; thereby this will be removed in final manuscript.

*- Section 3. It is not clear when you do the combined retrieval if you are doing the retrieval twice with the same measurements. Please explain clearly. If this is the case, care should be taken not to use the same measurements twice. Otherwise we will estimate a much smaller error than the real value. See comments about this above.*

In the revised manuscript we will improve the usage of the terminologies "simultaneous retrieval", "combined retrieval products" and "a posteriori corrected CH4 product". As aforementioned, we do a simultaneous retrieval of N2O and CH4 (only one retrieval process). Then we combine the N2O and CH4 products to compute the "a posteriori combined retrieval product.

*- Section 3.3 confusing - Eq 9 not well explained, probably because combined retrieval not well explained*

See comment above.

*- Section 4.2. usually time/space collocation windows are chosen with a criteria of little variability in this window. Is this the case here? Is this justified by any paper? If not, why is this particular collocation window chosen ? Reference?*

The spatial and temporal collocation criteria used here are based on previous studies that use HIPPO missions to validate space-based observations. Specifically, in the current work we have followed the works of Wecht et al. (2012) and Xiong et al. (2013), where each HIPPO vertical profile (covering ~220 km, 2.2 latitude, and ~20 minutes) is characterised by a mean location (latitude and longitude) and a mean time. Then, all the IASI observations within the box ±2° latitude/longitude centred at each HIPPO mean location and ±12h around every HIPPO mean profile are compared. Note that the 2°x2° validation box corresponds to the horizontal area where a HIPPO profile is typically measured.

*- Section 8 line 5 one before last paragraph. Again, combined retrieval seems to mean retrieving the same quatinty with the same measurements using oe. You will get a wrong error covariance matrix like this.*

See comment above.

**Anonymous Referee #4**

*Overall it is good and can be published after major revision. The major problem is that it is like a report, and many places should be rewritten to make it more concise. For example, the conclusion should be rewritten completely, and the first paragraph in section 8 should be removed.*

*For structure, sections 5,6,7,9 should be moved to the data or method part, and move the results in these sections to validation, or put some of them to a subsection in validation.*

*There are quite a lot grammar problems and need some efforts to polish the language.*

Following the referee's recommendation, the new validation results have been grouped and we have removed subsections such as section 2.5.

*References should include AIRS which has the similar CH4 and N2O products.*

The results of the following references of AIRS CH4 and N2O products have been included and discussed in the revised manuscript.

Xiong, X., E. S. Maddy, C. Barnet, A. Gambacorta, P. K. Patra, F. Sun, and M. Goldberg, Retrieval of nitrous oxide from Atmospheric Infrared Sounder: Characterization and validation, J. Geophys. Res. Atmos., 119, doi:10.1002/ 2013JD021406, 2014.
Xiong, X., Weng, F., Liu, Q., and Olsen, E.: Space-borne observation of methane from atmospheric infrared sounder version 6: validation and implications for data analysis, Atmos. Meas. Tech. Discuss., 8, 8563-8597, doi:10.5194/amtd-8-8563-2015, 2015.

*To say IASI as thermal "nadir" sensor is inappropriate. It scans and have a large swath.*

This has been corrected in the revised manuscript.

*The whole paper has not mentioned the quality control of the products, and is the 2X2 average use all retrievals ? I think the retrievals are made on for clear pixels, and it should be mentioned.*

Following the referee's comment, a description of the quality control of MUSICA/IASI retrievals has been included in the Section 2. They are filtered according to (i) the number of iterations at which the convergence is reached, (ii) the residues of the simulated–measured spectrum comparison, and (iii) cloud cover.

Indeed, the 2x2° averages are computed using all the IASI retrievals contained in this box.

*Why the HIPPO -2,-3,-4 data are not used in the validation? How many profiles have been used ?*

In the revised paper we use all HIPPO profiles from all 5 HIPPO missions with measurements covering at least the 2000-8000m altitude range. These are 441 different profiles.
We compare these profiles to all IASI measurements that have sufficient sensitivity for a layer around 4km altitude. For CH4 there are 145 different HIPPO profiles that have coincidences with at least one IASI measurement that has sufficient sensitivity. For N2O (IASI is less sensitive) there are 127 different profiles. Each HIPPO profile is compared to all IASI data with sufficient sensitivity in a 2°x2° area around the HIPPO measurements. There are many IASI data that fulfil these conditions, i.e. each HIPPO profile is compared to many individual IASI data.

In addition we compare IASI data that are sensitive for a layer around 10km altitude. Therefore, we require HIPPO profiles that reach at least 12.5km altitude. There are 22 different HIPPO profiles that we can use for a meaningful comparison to IASI CH4 and N2O retrievals at 10km. At 10km IASI N2O and CH4 data have a similar sensitivity.